# Methods for Assessing Oocyte Quality: A Review of Literature

**DOI:** 10.3390/biomedicines10092184

**Published:** 2022-09-04

**Authors:** Yassir Lemseffer, Marie-Emilie Terret, Clément Campillo, Elsa Labrune

**Affiliations:** 1Hospices Civils de Lyon, Service de Médecine de la Reproduction, 59 Bd. Pinel, 69500 Bron, France; 2Faculté de Médecine, Université Claude Bernard, Lyon 01, 8 Av. Rockefeller, 69008 Lyon, France; 3Center for Interdisciplinary Research in Biology (CIRB), College de France, CNRS, INSERM, Université PSL, 75006 Paris, France; 4LAMBE, Université d’Évry, CNRS, CEA, Université Paris-Saclay, 91025 Évry-Courcouronnes, France; 5INSERM U1208, Stem Cells and Brain Institute, 69500 Bron, France

**Keywords:** oocytes, quality, morphology, genetics, polarized light, follicular fluid, apoptosis

## Abstract

The rate of infertility continues to rise in the world for several reasons, including the age of conception and current lifestyle. We list in this paper potential non-invasive and invasive techniques to assess oocyte quality. We searched the database PubMed using the terms “oocytes AND quality AND evaluation”. In the first part, we study the morphological criteria, compartment by compartment, to then focus in a second part on more objective techniques such as genetics, molecular, apoptosis, or human follicular fluid that contain biologically active molecules. The main criteria used to assess oocyte quality are morphological; however, several other techniques have been studied in women to improve oocyte quality assessment, but most of them are invasive and not usable in routine.

## 1. Introduction

The rate of infertility continues to rise in the world for several reasons, including the age of conception and current lifestyle. Thus, the place of assisted reproductive technology (ART) grows every day. However, despite the development of ever more efficient and precise techniques, the success rate in ART remains at about 35% in terms of pregnancy per embryo transferred [1]. Since the quality of the oocyte determines the embryonic development potential, this unique and highly specialized cell takes an important place in research to improve ARTs results. Usually, oocyte quality is assessed visually by morphological criteria under a conventional microscope. The disadvantage of this method is that it is not objective because it is based on purely qualitative criteria, is operator-dependent, and requires years of training. Several studies have focused on the characterization and prediction of oocyte quality, with the aim of finding an objective, quantitative and reproducible method to evaluate the oocyte developmental potential and thus improve the success rates in ARTs in terms of pregnancy and live birth.

In this paper, we review the literature on potential non-invasive and invasive techniques to assess oocyte quality.

## 2. Materials and Methods

We searched the database PubMed using the terms “oocytes AND quality AND evaluation”. Eligible articles were selected first by the title, then a second screening using the abstract, and finally, the whole article. The following information was extracted from each study: reporting author, publication year, and method to assess human oocyte quality.

## 3. Morphological Criteria of Oocyte Quality

Usually, the methods used to assess oocyte quality are based on morphological analysis and classification of the oocyte according to each of its cellular compartments and surroundings. Thus, the cumulus cells, zona pellucida, polar body, perivitelline space, and cytoplasm are analyzed by conventional microscopy (Figure 1).

In this first part, we will review the morphological criteria and classifications used routinely today. In general, although these methods are still controversial due to their subjectivity, they provide important information for the selection of oocytes for intra cytoplasmic sperm injection (ICSI) and embryos to transfer [2].

### 3.1. Cumulus–Oocyte Complex (COC)

Cumulus–oocyte complex includes the oocyte and all the cells that surround it. COC are often classified according to their compactness and clarity. This clarity would be related to lipid accumulation in the ooplasm and may be associated with good developmental potential but not transferable between different species. In fact, it was shown only in bovine oocytes [2,3]. The presence of blood clots in the cumulus complex seems to be associated with decreased oocyte quality, fertility rates, cleavage rates, and blastocyst formation. Expanded, fluffy cumulus cells and expanded corona cells are associated with higher fertilization rates and pregnancy [1]. Cumulus cells are crucial for oocyte quality; they are at the base of the energy production by the COC and could protect the oocyte from reactive oxygen species [4]. Other studies show that oocyte quality increases when the oocyte is surrounded by more layers of cells [3]. There does not seem to be any correlation between COC morphology and fertilization or embryo cleavage; thus, this criterion seems to be useless for assessing oocyte quality in ICSI [5].

### 3.2. Polar Body

The polar globule is a small cell that is expelled during the maturation of the oocyte and contains 23 chromosomes. Several studies have shown that the integrity, shape, or size of the polar body could be used to assess oocyte quality and even be predictive of post-ovulatory aging status [2,4]. Fertilization rates and embryo quality are higher when the polar body is intact, with a correct shape and a smooth surface [1,3]. Oocytes with an intact polar body display higher fertilization rates, whereas oocytes with a large, irregular, or fragmented polar body give lower results [3,4,5]. The diversity of studies and their conclusions on the morphology of the polar body suggests that this is still a controversial criterion for assessing oocyte quality [5,6].

### 3.3. Zona Pellucida

The zona pellucida is an extracellular matrix that surrounds the oocyte. It is secreted by oocyte I at the pre-antral follicle stage, and the peri-oocyte cells of the corona radiate.

A thick zona pellucida or a thick inner layer of the zona pellucida is associated with better oocyte development and embryo quality [1]. However, another study concludes that the thickness of the zona pellucida has no influence on embryo development after ICSI [3]. Furthermore, a thinner zona pellucida may be associated with a better fertilization rate in connection with the facilitation of sperm penetration [1,3]. Finally, the study of the zona pellucida alone does not allow a correct evaluation of the oocyte quality; it is recommended to combine it with the study of other criteria.

### 3.4. Perivitelline Space

The perivitelline space is the space containing the polar globule, including between the zone pellucida and the membrane of the oocyte II. The perivitelline space exists prior to fertilization but plays an essential role after fertilization. Two criteria regarding the perivitelline space could be used to assess oocyte quality: size and content [1,3]. The different studies on these criteria emphasize that they are subjective and non-standardized but that it is preferable to use oocytes with a thin perivitelline space and devoid of granulation [1]. Another study suggests that the presence of granules in the perivitelline space may be related to gonadotropin impregnation and may therefore be a physiological phenomenon of maturation without any direct link with fertilization rates, cleavage, or embryonic quality [5]. Thus, it is not recommended to use the evaluation of the perivitelline space as a criterion of oocyte quality.

### 3.5. Cytoplasm

Several studies have analyzed the impact of morphological changes in oocyte cytoplasm on ICSI results in terms of fertilization rate, embryo quality, and pregnancy. Studies are contradictory regarding the link between cytoplasm changes and oocyte quality [2]. Some of the criteria put forward by these studies to determine oocyte quality are the presence of vacuoles, granulations, and inclusions. However, although taken together, these criteria could allow the selection of the highest potential oocyte, they are still too preliminary to be applied to oocyte selection [1,6]. Moreover, another study attempted to relate oocyte quality to the presence of smooth reticulum endoplasmic clusters, and their results show that the presence of those clusters could decrease the quality of the blastocyst [7]. Finally, only the severe defects in the cytoplasm should be considered abnormalities and could be used to assess oocyte quality in association with other criteria [5].

### 3.6. Morphometric Parameter

Oocyte shape is not known as a marker of quality, but there is a team that aimed to correlate mean oocyte diameter (MOD) with the quality of developing good-quality embryos. In fact, we did not find enough data to assess oocyte quality and fertilization rates by oocyte morphology, but this team showed that oocytes with MOD between 105.96 and 118.69 μm had higher rates of good day five blastocyst [8].

## 4. Other Criteria to Assess Oocyte Quality

Morphological criteria are the main tool used in ART protocols today to assess oocyte quality because of their non-invasive nature. However, several techniques have been developed to score the different oocyte compartments and surroundings and correlate them to oocyte quality. These techniques are quantitative and objective, which may facilitate their use in daily routine. However, they can be difficult to use and implement in clinical practice. In this second part, we will describe them compartment by compartment (Figure 1)

### 4.1. Follicular Fluid

Human follicular fluid (HFF) is collected during follicular puncture after stimulation, then placed in a petri dish for oocyte research. HFF contains biologically active molecules that play a role in follicular development. Therefore, its analysis could be interesting and represent a non-invasive (because the COC are isolated from de-liquid) and objective technique to evaluate oocyte quality. Some authors correlate oocyte quality to the expression of low molecular weight proteins belonging to the Insulin Growth Factor (IGF) family in the HHF. Higher concentrations of IGF binding protein 1 (IGFBP-1) were found in the follicular fluid surrounding mature oocytes. The same results were found for IGF1 [2]. Several studies have attempted to investigate the correlation between the expression of transforming growth factor-β (TGF-β) family markers in HFF serum and oocyte quality, but the results are not conclusive [2]. The level of zinc in HFF could be an interesting indicator of oocyte maturity. Indeed, a study shows that patients with a follicular zinc level <35 µg/mL have a lower number of mature oocytes [9]. Thus, some follicular fluid markers could be used as good and non-invasive criteria to assess human oocyte quality.

### 4.2. Cumulus–Oocyte Complex (COC)

Two approaches have been used in the literature to study cumulus cells: their rate of apoptosis and their genetic content.

To date, the importance of bidirectional communication between the oocyte and the follicular cells for their function and development is well established. Most studies support the idea that the degree of apoptosis of cumulus cells is negatively correlated with oocyte quality [2]. In fact, higher apoptosis rates increase oocyte immaturity and decrease fertilization [1]. A study shows that a higher number of cumulus cells undergoing apoptosis correlates with oocytes at the germinal vesicle stage or in metaphase I, meaning oocytes that are not mature (in metaphase II) [10]. In addition, Kyu Sup Lee et al. [11] show that the incidence of cumulus cell apoptosis can be used as a marker of oocyte quality, IVF outcome, and age-related decline in fertility. 

Analysis of the gene expression of cumulus cells could also identify new markers of oocyte quality and is the subject of a lot of research. It was shown that the expression of the gremlin gene plays an important role in oocyte quality, fertilization rates, and higher embryo quality by facilitating the luteinization of granulosa cells and allowing cumulus cell expansion [12]. Another study investigating the abundance of pentraxin 3 gene expression in cumulus cells showed that its level of expression was positively related to oocyte competence [2]. Another feature that has been studied in the literature is the relative length of the telomeres in the cumulus cells, showing that cumulus cells with a higher relative length of their telomeres correlate with mature oocytes and embryos of good quality [13]. Another study in women under the age of 38 showed that the expression of the following genes; hyaluronic acid synthase 2, follicle-stimulating hormone receptor, versican, and progesterone receptor allowed the selection of better quality oocytes [1]. Finally, the study of some pro-apoptotic (BAX, Caspase 8 and 3, and p53) and anti-apoptotic (Bcl 2 and BIRC5) genes in cumulus cells did not show any significant difference between mature and immature oocytes [14].

Thus, this method of assessing oocyte quality is objective but difficult to use in daily routines.

### 4.3. Cytoplasm

Several cytoplasmic markers have been studied in order to correlate them with oocyte quality. Some biochemical markers related to mitochondria and cellular energy consumption provide interesting results but use invasive techniques not applicable in daily clinical practice [2,4,8]. The mitochondria play an important role in oocyte maturation because this maturation needs an important level of ATP. Thus, a decrease in ATP and mitochondrial function leads to abnormalities of meiosis and non-expulsion of the polar globule. It was also shown that the localization and the structure of mitochondria were altered during IVM compared to fresh matured oocytes. The supplementation of IVM culture with antioxidants could increase the rates of maturation [15].

Moreover, cellular functions are mediated by intracellular temperature; its distribution within the cytoplasm could reflect the activity of organelles. A study using a fluorescent polymer thermometer (FPT) that diffuses into the cell showed a higher temperature in fresh matured oocytes and around the meiotic spindle compared to post-ovulatory aged oocytes [4]. It could be an interesting criterion for oocyte quality, but this technique is invasive and not suitable for daily practice. Thus, we should work on alternative and non-invasive methods for temperature measurement.

### 4.4. Meiotic Spindle

The meiotic spindle is fundamental for the alignment and separation of chromosomes during meiosis, hence its major interest in the study of oocyte quality. It is often studied in polarized light, which allows evaluation of its localization, shape, and refringence. Several studies have shown that oocytes with a birefringent spindle in polarized light lead to better embryonic development than those with a non-birefringent spindle [2,3]. Moreover, meiotic spindle length has been correlated with oocyte quality in some studies [2]. A decrease in the distance of the pericentriolar materials, which is the amorphous material that surrounds a pair of centrioles (microtubule organizing center), to the spindle pole would be a criterion for better oocyte quality and development [4]. Studies have shown that oocytes that do not show a meiotic spindle in polarized light were associated with a decreased rate of fertilization and blastocyst formation. A study compared polarized light microscopy to the gold standard of spindle evaluation, which is confocal microscopy (but which is an invasive method). This comparison showed that the non-visualization of the spindle is associated with meiosis abnormalities. This was confirmed by another study, but using oocytes matured in vitro [16]. This technique could be considered in clinical practice but is costly and requires significant experience.

Finally, the different techniques and criteria used or being explored are summarized in Table 1, with their benefits and limitations.

## 5. Discussion

This mini review of the literature confirms that the main criteria used to assess oocyte quality are morphological. Although used in daily practice, they remain subjective and operator-dependent. However, several other techniques have been studied in women to improve oocyte quality assessment by making it quantitative and objective; most of them are not applicable in daily clinical practice because they are invasive. The human oocyte is a rare, unique cell, which makes its study complicated. One of the limits of these different studies is the fact that we must work on immature oocytes, which would not be usable in daily clinical practice. Indeed, it would not be ethical to study mature oocytes in metaphase II because it would be a loss of a chance for the patients.

We limited this mini review to research in humans, but by extending the field to other mammals, it is possible to find new ways of exploring the human oocyte and especially metaphase II oocytes. For example, Raman spectroscopy was used to study oocytes in mice in a non-invasive manner. Raman spectroscopy is a promising tool that provides a unique spectral fingerprint of the cell in a non-invasive manner. However, it is a complicated technique to implement in daily practice due to its cost and complexity [17].

Terret and Campillo teams, working in mice, correlated oocyte stiffness (measured by the tension of the oocyte cortex) to aneuploidy. Oocyte stiffness is regulated by the evolution of actin and myosin networks. By modifying these networks, they engineered extra-soft oocytes by two different protocols and showed that this oocyte softening induces aneuploidy [18]. Thus, studying cortical tension could be promising and a new way to explore the human female gamete.

Furthermore, it would be conceivable to propose the idea of a time sequence of techniques that could be used in daily practice to refine the evaluation of oocyte quality before use in in vitro fertilization. This could allow us to avoid oocytes that would be immature and therefore responsible for fertilization, blastulation, and pregnancy failures. It would be useful to imagine a sequence that would be adequate in time in daily practice. In fact, the oocytes are extracted from the follicular fluid and then incubated at 37° for one hour before decoronization. They are then decoronized and observed under the microscope according to the techniques used to judge their maturity. The main objective morphological characteristic is the expulsion of the polar globule. Figure 2 could correspond to a time sequence usable in daily practice if the resources of the laboratory allow it, and if we suppose that, the time is respected between each step.

Finally, oocyte quality and their use in daily clinical practice is a major research issue at present because of the many failures of fertilization, early embryo development, and pregnancies during in vitro fertilization protocols. The difficulty of conducting research on this cell is mainly due to its precious and unique character, but also due to the difficulty of extrapolating animal results to humans and due to the fact that most human studies are conducted on immature oocytes and not on mature oocytes ready to be fertilized. Focusing on the latter point, in vitro maturation could be used to obtain and thus evaluate mature oocytes and compare them to non-mature ones. However, one could wonder if they will have the same characteristics as in vivo matured oocytes.

## Figures and Tables

**Figure 1 biomedicines-10-02184-f001:**
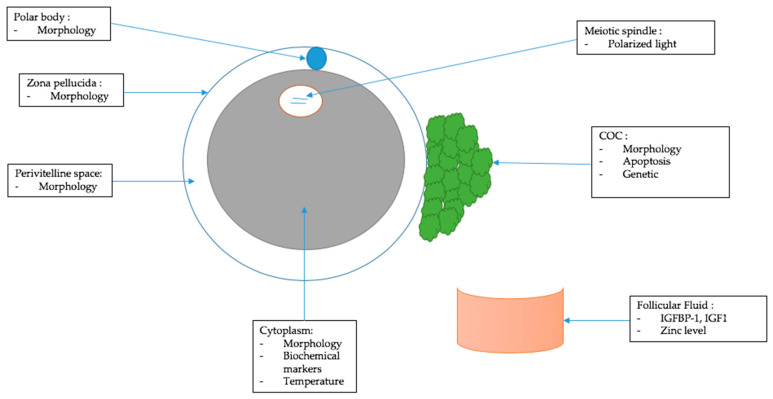
Criteria used or studied to assess oocyte quality.

**Figure 2 biomedicines-10-02184-f002:**
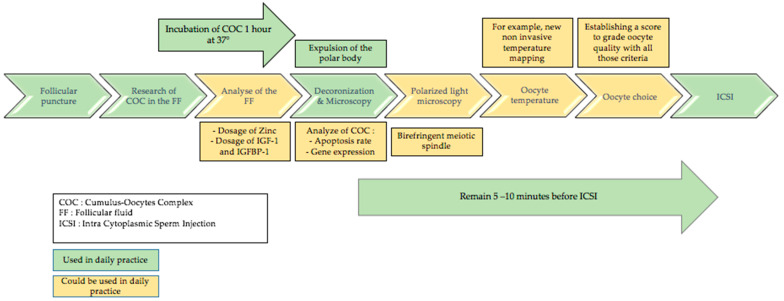
Time sequence to assess oocyte quality in daily practice.

**Table 1 biomedicines-10-02184-t001:** Criteria used or studied to assess oocyte quality.

	Criteria	Outcome	Strength	Weakness
**COC**	Compactness and clarity [2,3]	↗ developmental potential	Non-invasive	Subjective Bovine oocyte
	Presence of blood clots [1]	↘ oocyte quality; fertility, cleavage, and blastulation rates	Non-invasive Easy to analyze under microscope	Not reproducible
	Expanded corona cells [1]	↗ fertilization rates and pregnancy	Non-invasive	Subjective Not reproducible
	Number of layers of cells [3]	↗ oocyte quality	Non-invasive	Not quantifiable and not reproducible
	Higher apoptosis rate [1,3,9]	↘ oocyte quality; fertilization	Objective Reproducible	Difficult to use in daily routine
	Gene expression of gremlin [2]	↗ oocyte quality, fertilization rates, and embryo of good quality	Objective Reproducible	Difficult to use in daily routine
	Higher length of telomeres [10]	↗ oocyte quality, embryo of good quality	Objective Reproducible	Difficult to use in daily routine
	Expression of pro and anti-apoptotic genes	No significant difference	None	Useless
**Polar body**	Integrity, correct shape, and size [2,3,4,5,6]	↗ fertilization rates ↗ embryo quality ↗ oocyte quality	Non-invasive Easy to analyze under microscope	Subjective Controversial
**Zona Pellucida**	Thick zona pellucida [1,3]	Better oocyte development ↗ fertilization rates	Non-invasive	Subjective Controversial
**Perivitelline space**	Size and content [1,3]	No link found	None	Subjective Non-standardized
**Cytoplasm**	Vacuoles, granulations, and inclusions [1,2,5,6]	No link found	None	Contradictory studies Preliminary
	Biochemical markers [2,4,8]	Oocyte quality	Could be objective	Not applicable in daily routine Invasive
	Intracellular temperature [4]	Higher temperature in fresh mature oocyte	Objective	Invasive Not applicable in daily routine
**Morphometric**	MOD	↗ quality of day 5 blastocyst	Non-invasive Easy to use with time laps	Does not help to choose the oocyte
**Follicular fluid**	Higher level IGFBP-1 and IGF1 [2]	Mature oocyte	Non-invasive	Needs to be standardized
	Level of zinc < 35 µg/mL [7]	Lower number of mature oocyte	Non-invasive Reproducible	Needs to be standardized
**Meiotic spindle**	Birefringent [2,3]	Better embryonic development	Non-invasive Reproducible	Costly Experience
	No MS seen in polarized light [2]	↘ rate of fertilization and blastocyst formation	Non-invasive Reproducible	Costly experience

## Data Availability

Not applicable.

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
