# Peer review of "Methods for Assessing Oocyte Quality: A Review of Literature"

_biomedicines, 2022, doi:10.3390/biomedicines10092184_

Round 1

Reviewer 1 Report

In this concise manuscript, the authors reviewed key literatures regarding major non-invasive and invasive techniques for assessing human oocyte quality. For the non-invasive techniques, the authors further elaborated the morphological approach with brief descriptions and discussions on observations from relevant cells and structures. For the invasive techniques, the authors provided specific and concise examples of respective methods. The useful compilation uncovered in this manuscript thus provides inspiring insights for both the scientific community and the practitioners.

However, the author should consider addressing the following issues in this manuscript:

Major points:

1. It would be better if the authors could formulate one table summarizing key information from the selected literature such as techniques, criteria, special notes, strength & weakness of the respective techniques, publication year, etc. And by comparing the relative strength and weakness of each major technique, in the main text, when possible, the authors might consider adding brief descriptions of how these techniques/criteria evolved in this field from a perspective of science & technology history.

2. It would be ideal if the authors could provide a figure depicting the suggested/optimized workflow of accessing oocyte quality using current mainstream techniques and criteria.

3. The author should consider providing a more comprehensive outlook for new methods in evaluating human oocyte quality, proposing possible directions and associated technical challenges, e.g., whether it is possible to directly incorporate ER(endoplasmic reticulum)- or lysosome/autophagosome-related markers for the quality assessment of oocyte and associated cells.

Minor points:

1. To make the text easier for others to understand, the author might consider making more precise and detailed statements, e.g., line 142-144, "It was shown that the expression of the gremlin gene could be a predictive marker of three outcomes: oocyte maturity, fertilization and embryonic quality" is too general.

To summarize, the topic of the review is important and the paper may be relevant for Biomedicines. The manuscript covers key discoveries in this field and could be improved by addressing the above issues, I therefore recommend reconsideration of the manuscript after major revision.

Reviewer 2 Report

Please the Authors to refer to the attached file

Round 2

Reviewer 1 Report

In this version, the authors addressed several issues mentioned in the previous comments, including:

Major points:

1:  incorporated a table summarizing current methods for assessing oocyte quality;

3.  added in new ER-related perspectives for quality assessment;

Minor points:

1. substituted the original text with a more clear description of the principle involved in this particular evaluation.

Yet, for major point 2, the authors provided a figure generally depicting the current methods for oocyte quality assessment which partially fills the gap. The authors should discuss in the text, whether and how in practice, combing several non-invasive methods for oocytes or combing non-invasive methods for oocytes & invasive methods for COC, could give a better quality assessment.

To summarize, the authors have made several reasonable modifications to the previous version and the manuscript could be improved by addressing the remaining issues stated above, I therefore recommend acceptance of the manuscript after minor revision.

Author Response

Thank you again for this comment, very useful. We understand better what you expected from us on this point. We have therefore included a paragraph and a figure in the discussion (line 235) responding to your comment. We remain at your service for any further comments.